# Is Strain Elastography Useful in Diagnosing Chronic Autoimmune Thyroiditis in Children?

Cristina Mihaela Cepeha [1], Corina Paul [2,*], Andreea Borlea [3], Renata Bende [3], Monica Simina Mihuta [1] and Dana Stoian [3]

1   PhD School Department, Victor Babes University of Medicine and Pharmacy, 300041 Timisoara, Romania
2   Department of Pediatrics, Victor Babes University of Medicine and Pharmacy, 300041 Timisoara, Romania
3   Department of Internal Medicine II, Victor Babes University of Medicine and Pharmacy, 300041 Timisoara, Romania
*   Correspondence: paul.corina@umft.ro

**Abstract:** Strain elastography (SE) determines the level of tissue stiffness and thus helps in the diagnosis and differentiation of various pathologies. This paper aims to study the elastographic differences between healthy thyroid parenchyma and ones affected by chronic autoimmune thyroiditis (CAT) in children. We included in our study 52 children diagnosed with CAT and 22 children with no thyroid pathology. After clinic examination and biological tests, an ultrasound was performed followed by SE using a Hitachi Preirus machine with a 5–15 multi-frequency linear probe. The mean strain ratio (SR) values were significantly lower in the control group compared with the CAT group ($0.68 \pm 0.2$ vs. $1.19 \pm 0.25$; $p < 0.0001$). A mean value above 0.9 was found predictive for CAT with 84.62% Sensitivity(Se), 95.45% Specificity (Sp), 97.8% Positive Predictive Value (PPV), 72.4% Negative Predictive Value (NPV), and area under receiver operating characteristic (AUROC), 0.9. No differences were found between the two thyroid lobes. Also, we found no differences between girls and boys. Out of the 52 children with CAT, 39 were on therapy with levothyroxine. No differences were found between SR values in preexisting hypothyroid cases compared to euthyroid cases ($1.24 \pm 0.26$ vs. $1.18 \pm 0.25$; $p = 0.4615$). Our results show that SE is a useful examination technique of children diagnosed with CAT, in accordance with other studies conducted on adults. This study lays the foundation for elastographic examination among children.

**Keywords:** strain; elastography; chronic autoimmune thyroiditis; thyroid; children

## 1. Introduction

The two major categories of autoimmune thyroid disease (AITD) are Graves's disease (GD) and chronic autoimmune thyroiditis (CAT) or Hashimoto's thyroiditis (HT). Both are characterized by an autoimmune system disorder leading to an autoimmune attack against thyroid tissue [1]. CAT is the most common AITD, and it is considered to be the most common autoimmune disease, with an incidence of 20–30% and increasing, depending on the population group studied [2]. Up to 27% of women are diagnosed with this condition, the incidence increasing after the age of 50, while males are affected only up to 7% [3]. It is, as well, the most common cause of hypothyroidism [4,5]. HT is also the most common autoimmune pathology in the pediatric population. The prevalence among children is reported to be 4–10% [6,7]. Etiopathogenesis is not completely defined. In most cases, primary CAT occurs, without identifiable causes. Six forms of primary HT were identified: juvenile form [8], classical form, fibrous variant [9], IgG-4 related [10,11], Hashitoxicosis, and silent (painless) form that occurs sporadically or postpartum [12,13]. Although thyroid lymphoma is extremely rare, persons with Hashimoto's thyroiditis have a greater risk of developing the condition. In thyroid glands resected for a tumor, the lymphocytic infiltrate of HT is usually seen. HT patients have a much higher frequency of

primary thyroid lymphomas, which strongly implies a pathogenetic connection between this autoimmune condition and malignant thyroid lymphoma [14]. Although CAT is the thyroid condition most closely linked to chronic hypothyroidism, any type of thyroiditis can lead to lifelong hypothyroidism. The diagnosis is based on the clinical manifestations, the presence of anti-thyroid antibodies in the peripheral blood [anti-thyroid peroxidase (ATPO) and antithyroglobulin antibodies (ATG)], and the ultrasound appearance. Clinical manifestations are absent most of the time. When they are present, they are most often due to hypothyroidism (constipation, dry skin, cold intolerance, hair loss, weight gain, myalgias, etc.), and sometimes they are symptoms caused by the compression of the enlarged thyroid gland on neighboring structures (dyspnea, dysphonia, dysphagia) [15]. Children and infants present more often with lethargy and failure to thrive. Thyroid ultrasound (US) is an important evaluation tool, providing valuable information about the thyroid volume, echogenicity and homogeneity of the gland. Usually, the thyroid affected by CAT shows a hypoechoic, inhomogeneous image due to follicular destruction, with variable volume. Apseudolobulated appearance may be found in some cases due to fine echogenic fibrous septae. In some cases, Color Doppler may indicate a slightly increased vascularity [16,17]. The treatment is mostly medical, surgery is chosen as a means of treatment in cases with compressive cervical symptoms, for aesthetic reasons, or when a patient has a hard-to-differentiate nodular lesion [18,19]. The treatment of hypothyroidism is carried out by daily and lifelong intake of synthetic levothyroxine orally [20,21]. For infants and kids, the dosage of levothyroxine is weight-based, varies with age and should be adjusted based on clinical response and biochemical parameters.Patients with initial TSH values over10 mIU/L, patients with symptoms of hypothyroidism and TSH levels between 5 and 10 mIU/L, and patients who are pregnant or trying to conceive should all be treated with levothyroxine. Combination triiodothyronine/thyroxine therapy is not advised and offers no benefits over thyroxine monotherapy [22]. Thyrotropin-suppressing doses of levothyroxine can be administered over the short term to patients with HT and a large goiter in order to reduce the size of the goiter. After six months of treatment with levothyroxine, the size of the goiter will typically decrease by 30%, regardless of whether they have a euthyroid or hypothyroid condition [23]. For the IgG4-related type, a treatment option is glucocorticoids which have been shown to help cure the disease and can avoid the need for levothyroxine treatment [24].

As an additional method to ultrasonography, elastography has been developed in recent years. There are two main categories: strain imaging (strain elastography (SE) and acoustic radiation force impulse (ARFI)) and shear-wave imaging (SWI). There are three technical approaches for SWI: 1-dimensional transient elastography (1D-TE), point shear wave elastography (pSWE), and 2-dimensional shear wave elastography (2D-SWE) [25–27]. Both SE and shear-wave elastography (SWE) have been studied on the liver [28,29], kidney [30], spleen [31], prostate [32], breast [33,34], muscle [35], thyroid [36], parathyroid glands [37], etc. SE measures the tissue deformation generated by applying external pressure while SWE measures the speed of shear waves generated by the machine [38]. SE was the first technique that appeared. A slight force is applied either by the examiner, or the transducer is held still, and the movement of the aorta is used for compression. With no moving along the transducer's long axis or perpendicular to the long axis, the compression should be uniform and as near to vertical to the skin as possible. Elastograms produced by sliding movements are noisy and of poor quality. As greater pressure is applied, the soft tissues will appear stiffer because biological tissue deforms non-linearly. As a result, there should be minimal pre-compression and lots of gel application with the transducer held gently barely in contact with the skin. Thetissuedisplacementisdeterminedfrom the received high-frequency data (RF)bycomparingsuccessive frames usingthesoftwareintercorrelation method. The resulting strain data is graphically displayed as a two-dimensional map called an elastogram. Elastograms are typically displayed as translucent overlays on grayscale images, using color-coded scales that can vary between US systems [39]. Some

users prefer to use red to signify stiffness (indicating danger or alarm) and blue to signify softness, while others have chosen the opposite.

Given the importance of malignant lesions, most elastographic studies of thyroid tissue have focused on nodular pathology and the distinction between malignant and benign nodules. SE proved to be useful in the evaluation of thyroid nodules being an additional method, adjunct to ultrasonography in differentiating malignant from benign nodules [40–43]. The EFSUMB guidelines on SE state that the method may be considered, in the hands of specialists, as a beneficial addition to US, increasing its accuracy for thyroid cancer identification [44]. It was found that SE can be used with good specificity and high sensitivity to detect malignant thyroid nodules [45]. Every elastographic technique has the common limitation that requires experience to obtain repeatable and accurate elasticity readings. Pre-stress can generate misleadingly high stiffness values, especially in superficial tissues. The fibrotic-atrophic involution that occurs in all types of thyroid nodules, whether benign or malignant, is a significant general constraint in the measurement of thyroid nodule elasticity. The presence of carotid artery pulsations in the thyroid gland, which causes compression-decompression movementsis a special limitation of thyroidelastography [46]. Non-invasive techniques are preferable, especially when frequent evaluations are required. One such efficient and trustworthy technique for evaluating the thyroid in adults is elastography. Children, however, are different from adults. The measurement reproducibility and diagnostic accuracy in the pediatric population can be influenced by several variables, including age and breath holding.

There are also some studies on the usefulness of strain elastography in the differentiation of autoimmune thyroid diseases [47,48], but on smaller groups of patients, leaving the possibility of new research directions.

Although important, CAT has not been sufficiently studied in the pediatric population. Thus, our study proposes the elastographic evaluation of thyroid parenchyma affected by CAT in children. We aim to determine if there are significant differences between the level of elasticity of the thyroid parenchyma in children affected by CAT compared to healthy ones, as well as establish a cut-off value for the diagnosis of HT.

## 2. Materials and Methods

### 2.1. Group Characteristics

This prospective study took place between January and July 2022. It was conducted following the Ethics Guidelines of the Helsinki Declaration, and the local ethics committee approved it.

The parents of each child participating in the study completed an informed consent form. Fifty-two children (14 boys, 38 girls, aged 7–18) diagnosed with CAT were included along with 22 children (5 boys and 17 girls, aged between 6–18) with no thyroid pathology.

### 2.2. Inclusion Criteria

We included in our study children with chronic autoimmune thyroiditis diagnosed based on clinical examination, ultrasound appearance and elevated levels of ATPO and ATG antibodies. Out of 52 children with CAT, 39 were on hormone replacement therapy. Also, for the control group, children without thyroid pathology, with normal laboratory tests and normal thyroid ultrasound appearance were included. All children were examined at " Dr. D" Medical Center in Timisoara, Romania.

### 2.3. Exclusion Criteria

Patients with nodular thyroid pathology, malignancies, or a history of thyroid surgery were excluded from the study. Patients with Graves' Disease (GD) or those with acute or subacute thyroiditis were also not included in the study. Additionally, because of the challenging assessment, children under the age of 6 were excluded from the study. Cases that had suggestive ultrasonography findings for CAT but normal antithyroid antibody titers were also not included.

### 2.4. Biochemical Assay

For each child, the following parameters were analyzed: TSH—thyroid-stimulating hormone (reference range 0.6–4.84 µIU/mL; ECLIA method—immunochemistry with enzyme chemiluminescence immunoassay), FT4—free-thyroxine (reference range 12.5–21 pmol/L; ECLIA method), ATG (reference range <37 IU/mL; ECLIA method), and ATPO (reference range <26 IU/mL; CMIA method—microparticle-based chemiluminescence immunochemistry). All biological determinations were carried out in an accredited laboratory.

### 2.5. Conventional Ultrasound and Elastography Examination

A Hitachi Preirus machine with a 5–15 multi-frequency linear probe was used to perform Conventional B-mode ultrasound and strain elastography (real-time elastography—RTE) of the thyroid. All subjects were clinically evaluated (including thyroid palpation and height and weight measurements), and then grey-scale thyroid ultrasonography was performed. The optimal examination position is in the supine position, with the head tilted back to expose the neck. The child is then asked not to talk for a few seconds and not to swallow. Transverse diameters (two dimensions) and a longitudinal diameter (one dimension) were measured, the thyroid volume being calculated by the device and expressed in milliliters (mL).Using the probe perpendicular to the skin, slight, repetitive compressions were applied. All images were obtained in the longitudinal plane. The color blue signifiesno strain (high stiffness), green suggestsintermediate stiffness, and red indicatessoft tissue on a blue-green-red color map. Two regions of interest (ROI) were positioned forstrain ratio (SR) calculation. The thyroid tissue was represented by ROI A, while ROI B represented the sternocleidomastoid muscle in front of the ipsilateral thyroid parenchyma.Avoiding areas of thick calcification is advised because these regions tend to be stiffer and may not be an accurate representation of the surrounding soft tissue. Similar to this, regions of interest should not be positioned posterior to cystic areas because of the probable artifactual increased stiffness.Each lobe had five successive measurements, and the mean value was considered for analysis. For each lobe, the SR was calculated and automatically presented (Figures 1 and 2).

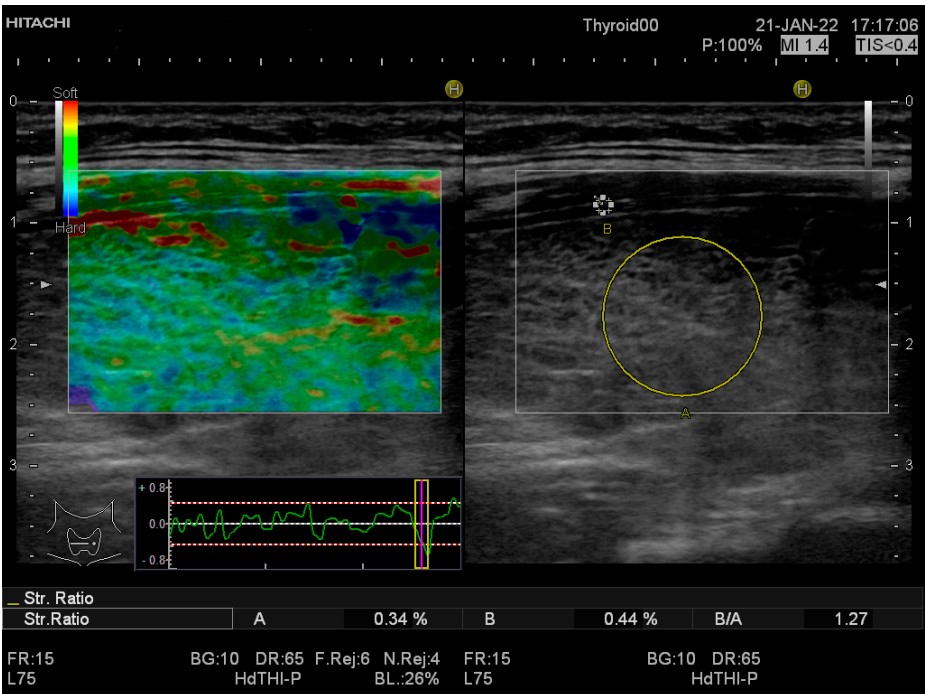

**Figure 1.** Strain elastography (**left**) and conventional US (**right**) of a girl, age 17, diagnosed with CAT; The yellow circle A is the local ROI on the thyroid tissue; the yellow circle B is the local ROI on the adjacent sternocleidomastoid muscle. SR = 1.27.

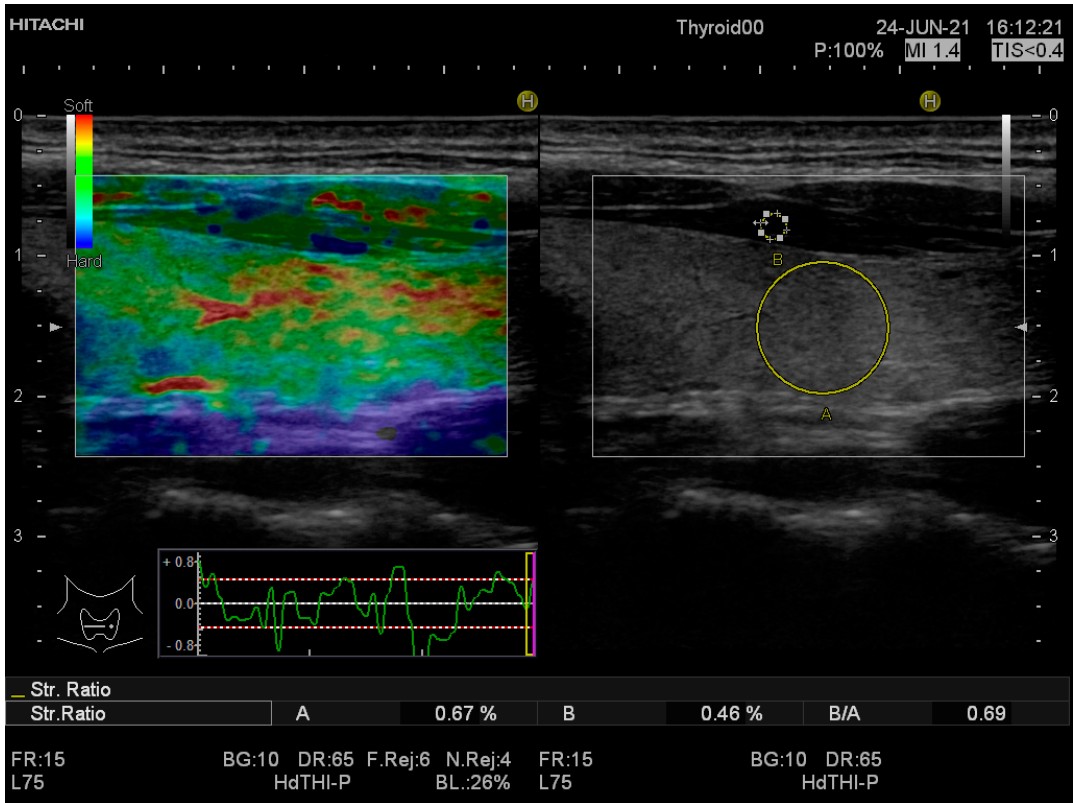

**Figure 2.** Strain elastography (**left**) and grey-scale ultrasound (US) (**right**) of a girl, age 16 with no thyroid pathology. The yellow circle A is the local region of interest (ROI) on the thyroid tissue; the yellow circle B is the local ROI on the sternocleidomastoid muscle adjacent to the thyroid. Strain ratio (SR) = 0.69.

*2.6. Statistical Analysis*

The statistical analysis was performed using MedCalc Version 19.4 and Microsoft Office Excel 2019 (Microsoft for Windows). Descriptive statistics were used for demographic, anthropometric, and laboratory findings. Kolmogorov Smirnov test was used for testing the distribution of numerical variables. The student's t-test was used for group comparisons of continuous variables with a normal distribution, and non-parametric tests (Mann Whitney U-test) were applied for variables with a non-normal distribution. Group comparisons of categorical variables were performed using Pearson's $x^2$-test. Linear regression analysis was used to evaluate the correlation between mean SR values and several prediction factors. The individual impact of different prediction factors on the variance of continuous variables was assessed by building multivariate regression models. The predictors, in the final regression equations, were accepted according to a repeated backward-stepwise algorithm (inclusion criteria $p < 0.05$, exclusion criteria $p > 0.10$) in order to obtain the most appropriate theoretical model to fit the collected data. For each predictive test, 95% confidence intervals (CI) were calculated, and a *p*-value below 0.05 was considered to concede statistical significance.

## 3. Results

Thyroid elastography measurements using strain elastography were performed on 74 subjects, 52/74 (70.3%) diagnosed with CAT, and 22/74 (29.7%) without thyroid pathology. A total of 22 (29.8%) subjects had a family history of CAT, 3 (4%) had other autoimmune pathology, and 7 (9.5%) of non-autoimmune thyroid disease. Reliable measurements were obtained in 74/74 subjects (100%). In both groups, the percentage of the female gender was over 70%, compared to the male gender, under 30%. There were no differences between the two groups regarding weight or height. The mean TSH value for the CAT group was

2.65 ± 1.17, while, for the control group, it was 3.28 ± 1.08 ($p$ = 0.0901). The mean Ft4 value for healthy children was 13.58 ± 1.56, while, for the CAT group, it was 14.20 ± 1.53 ($p$ = 0.1175). The main characteristics of the subjects included are summarized in Table 1.

**Table 1.** Main characteristics of the study group.

| Parameter | Children with CAT $n$ = 52 | Healthy Children $n$ = 22 | $p$ Value |
|---|---|---|---|
| Age (years) | 12.66 ± 3.31 | 12.77 ± 2.92 | 0.8929 |
| Gender (%) Female Male | 73.1% (38/52) 26.9% (14/52) | 77.3% (17/22) 22.7% (5/22) | 0.9307 0.9307 |
| Weight (kg) | 48.35 ± 18.65 | 41.31 ± 7.22 | 0.0913 |
| Height (cm) | 149.73 ± 16.49 | 146.72 ± 10.73 | 0.4339 |
| BSA (m$^2$) | 1.40 ± 0.33 | 1.29 ± 0.15 | 0.1393 |
| TSH | 2.65± 1.17 | 3.28 ± 1.08 | 0.0901 |
| FT4 | 14.20 ± 1.53 | 13.58 ± 1.56 | 0.1175 |
| ATPO | 400 [24–1450] | 11 [2.9–36] | <0.0001 |
| ATG | 125.5 [1.3–2380] | 14.6 [2.9–45] | <0.0001 |
| Thyroid volume (Mean ± SD) | 14.71 ± 6.42 | 9.04 ± 3.03 | 0.0002 |

CAT—chronic autoimmune thyroiditis; BSA—Body Surface Area; TSH—thyroid stimulating hormone; FT4—free thyroxine; ATPO—Anti-thyroid peroxidase antibodies; ATG—Antithyroglobulin antibodies; SR—Strain Ratio; SR-LTL—Left thyroid lobe SR; SR-RTL—Right thyroid lobe SR.

SR was performed on both thyroid lobes, and no differences were found between the mean values obtained in the left lobe compared to the right lobe for the subjects included in the CAT subgroup (1.20 ± 0.31 vs. 1.18 ± 0.27; $p$ = 0.7264), nor in healthy subjects (0.66 ± 0.19 vs. 0.70 ± 0.24; $p$ = 0.5432), as presented in Figure 3.

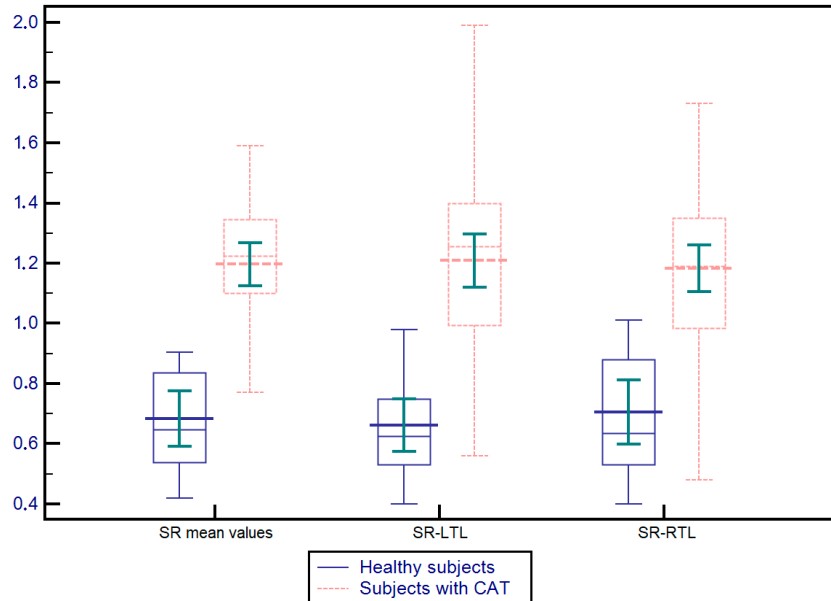

**Figure 3.** Box-and-whisker distribution plots comparing SR mean values, SR-LTL, and SR-RTL, respectively in the two subgroups (healthy subjects vs. subjects with CAT).

The mean thyroid stiffness (TS) values were significantly higher for children with CAT compared to the healthy controls (1.19 ± 0.25 vs. 0.68 ± 0.2; $p$ < 0.0001) (Table 2).

In children with CAT, no difference between the mean SR values related to gender were observed (1.19 ± 0.24 for females vs. 1.20 ± 0.30 for males; *p* = 0.9014).

**Table 2.** Mean SR values in the two groups of children.

| Parameter | Children with CAT *n* = 52 | Healthy Children *n* = 22 | *p* Value |
|---|---|---|---|
| SR mean value | 1.19 ± 0.25 | 0.68 ± 0.2 | <0.0001 |
| SR-LTL | 1.20 ± 0.31 | 0.66 ± 0.19 | <0.0001 |
| SR-RTL | 1.18 ± 0.27 | 0.70 ± 0.24 | <0.0001 |

CAT—chronic autoimmune thyroiditis; SR—Strain Ratio; SR-LTL—Left thyroid lobe SR; SR-RTL—Right thyroid lobe SR.

The optimal cut-off value defined as the highest sum of sensitivity (Se) and specificity (Sp) determined using the mean RS values for predicting the presence of CAT in children was >0.9 (AUROC—0.90, Se—84.62%, CI 95%: 71.9–93.2; Sp—95.45%, CI 95%: 77.2–99.9; PPV—97.8% and NPV—72.4%) as illustrated in Figure 4.

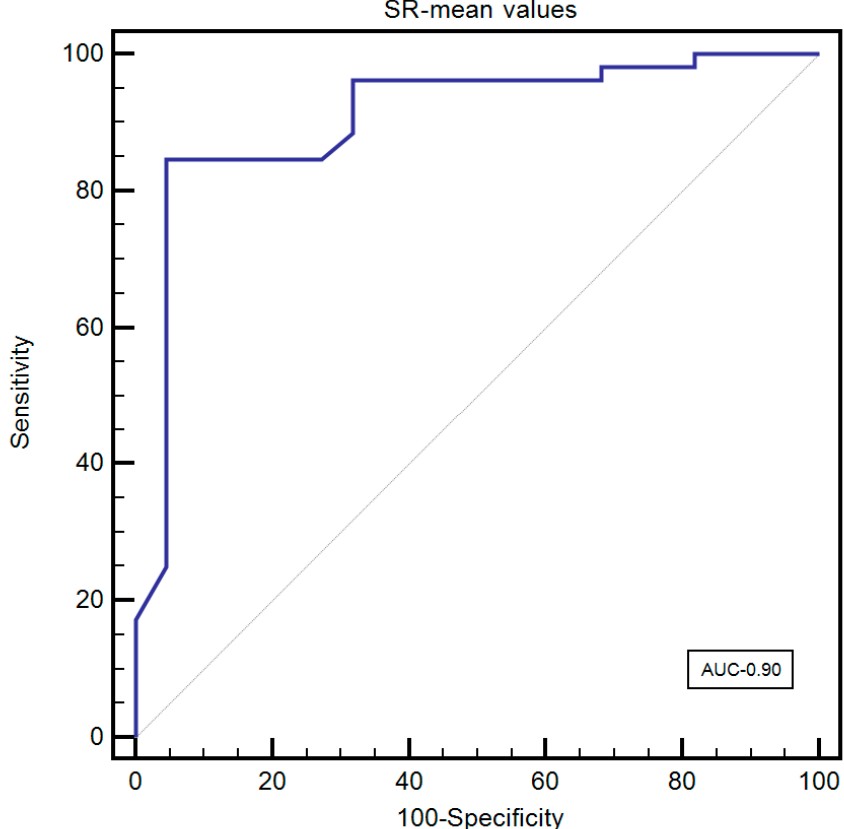

**Figure 4.** The area under the receiver operating characteristic curve (ROC-curve) for SR measurements.

Regarding substitution therapy, 75% (39/52) of the children in the CAT group were receiving LT 4replacementtherapy. No differences were found between mean SR values in preexisting hypothyroid cases compared to euthyroid cases (1.24 ± 0.26 vs. 1.18 ± 0.25; *p* = 0.4615) (Figure 5). In the control group, no child needed treatment.

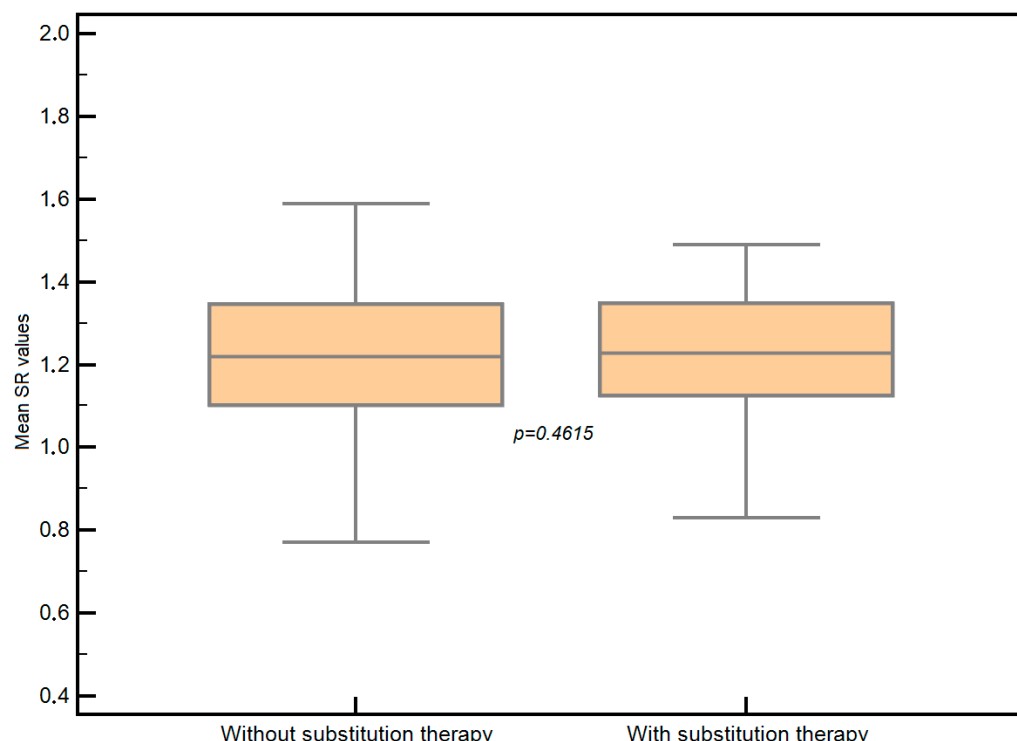

**Figure 5.** Box-and-whisker distribution plots comparing SR mean values, in euthyroid patients compared to treated hypothyroid patients.

In univariate regression analysis, the following parameters were associated with the presence of CAT: weight ($p = 0.015$), height ($p = 0.031$), BSA ($p = 0.005$), thyroid values ($p < 0.001$), ATPO antibody levels ($p < 0.001$), ATG antibody levels ($p = 0.017$), and family history of CAT ($p < 0.001$).Multivariate regression analysis was used to evaluate the independent factors associated with CAT. The regression model was built based on the forward stepwise method and Akaike information criteria (AIC) was used to appreciate the best model. The model including thyroid volume ($p = 0.015$), family history of CAT $p = 0.013$), and ATPO antibody levels ($p = 0.003$) was associated with the presence of CAT in children.

A positive and significant correlation was found between SR mean values and ATPO levels (r = 0.65, $p < 0.0001$), thyroid volume (r = 0.49, $p < 0.001$), body surface area (r = 0.32, $p = 0.0054$), and ATG levels (r = 0.29, $p = 0.0135$), respectively (Table 3).

**Table 3.** Correlation between SR and different parameters.

| Parameters | r Coefficient | Correlation |
|:---:|:---:|:---:|
| ATPO | 0.65 | Strong positive |
| ATG | 0.29 | Weak positive |
| Thyroid volume | 0.49 | Strong positive |
| BSA | 0.32 | Moderate positive |

ATPO—Anti-thyroid peroxidase antibodies; ATG—Antithyroglobulin antibodies; SR—Strain Ratio; BSA—Body Surface Area; r—Pearson's coefficient for correlation.

No significant correlation was found between SR mean values and TSH values ($p = 0.8205$), FT4 values ($p = 0.8232$), or age ($p = 0.0840$).

## 4. Discussion

Conventional ultrasound proved its usefulness in examining the thyroid over a period of time. Although it only suggests the diagnosis, as biological determinations are also

necessary, it is highly suggestive for CAT if the appearance ishypoechoic and heterogeneous. It can also have micronodules or a pseudolobulated appearance of the parenchyma [16]. Several studies have shown that elastography is an adjuvant method, adding value to the classic examination.

SE was studied on various groups of patients. A study conducted in Baghdad that included 25 patients with diffuse thyroid disease (DTD) and 25 control subjects found significant differences between the two groups. The mean value of elasticity for the patient group was 1.36, much higher than the control group, 0.82. The cut-off value established for AITD was 0.89 (80% Sensitivity, 70% specificity) [49]. Another study compared the thyroid stiffness of patients with CAT and healthy controls with significant differences between the groups. The median SR value for CAT group was $1.39 \pm 0.72$, while the control group had much lower values ($0.76 \pm 0.55$). The optimal cut-off value for the prediction of CAT was 0.677 (96% sensitivity, 67% specificity, AUROC 0.775). In addition, a positive correlation was found between SR and the values of ATPO ($r = 0.682$) [50]. Korkmaz et al. included in their study 94 patients with HT and 82 subjects with no thyroid pathology. Although they obtained differences between the study groups ($8.4 \pm 9.6$ vs. $1.37 \pm 0.8$), the values were higher than those obtained in the two studies mentioned previously [51]. A study that included 180 CAT patients and 70 healthy subjects established the optimal cut-off value as 1.64 (69% sensitivity, 92% specificity, 95,4 PPV, 54% NPV, AUROC 0.87). The mean SR values for CAT group were $2.81 \pm 2.11$, significantly higher than the control group ($1.03 \pm 0.51$) ($p < 0.0001$). No statistically significant correlation was found between thyroid stiffness and ATPO [52]. Although different mean values and cut-off values were obtained, all the previously mentioned studies concluded that there are significant stiffness differences between the healthy thyroid parenchyma and the one affected by CAT when using SE.

Compared to studies on adult population, studies that include children are fewer. There are some studies that compare thyroid elasticity in children using the SWE method [53,54]. The cut-off values proposed for diagnosing CAT varies: 1.41 m/s [55], 2.39 m/s [53], 12.2 kPa [54], as well as the mean values: $3.7 \pm 1.2$ (2.59–6.25) m/s [53], 14.9 kPa (12.9–17.8 kPa) [56]. However, significant differences between the thyroid elasticity of healthy children compared to children diagnosed with HT were observed in all papers mentioned above.

To our knowledge, there are only two studies that include children with CAT evaluated using SE.Yurttan et al. conducted a study on 54 healthy children to determine the SR of normal thyroid parenchyma. The mean value found was $0.54 \pm 0.38$, results that are consistent with our results ($0.68 \pm 0.2$). No correlations were found between SR and age ($r = 0.22$; $p = 0.15$) or gender ($r = 0.007$; $p = 0.96$) [57].

A study conducted on 63 children diagnosed with HT and 47 children without thyroid pathology revealed differences between them ($1.75 \pm 1.46$ vs. $0.26 \pm 0.77$; $p < 0.001$), results similar to our results. Instead, the cut-off value established for the presence of CAT was 0.31 (92.1% Se, 66% Sp, AUROC 0.828), much lower than the cut-off value we obtained (>0.9). They also found no correlation between SR and TSH, but a positive correlation between SR and ATPO [58].

Another study that included 76 adolescents with HT and 46 without thyroid pathology recommends the cut-off value for CAT diagnosis >0.98 (83% Se; 93% Sp; AUROC 0.929), a result in accordance with the cut-off value we obtained (>0.9; AUROC 0.9, Se 84.62%, Sp 95.45%; PPV 97.8%; NPV 72.4%). The mean values for children with CAT were $1.2 \pm 0.2$, significantly higher than the mean SR values for control subjects $-0.77 \pm 0.18$ ($p < 0.01$), results that are similar to our results [59].

Regarding correlations, we found a strong positive correlation between SR and ATPO values ($r = 0.65$, $p < 0.0001$) and a weak positive correlation between SR and ATG values ($r = 0.29$, $p = 0.0135$). In accordance with our results, there were also the results obtained by Ozturk et al. who also obtained positive correlations between SR and ATPO ($r = 6.85$), but the possible correlation between SR and ATG was not analyzed [58]. Çekiç et al. also

found a correlation between SR and ATPO (r = 0.439, $p < 0.01$) but no correlation between SR and ATG [59]. We did not find correlations between SR and TSH, in accordance with the literature [58]. No correlations between SR and ft4 values were found. We also found correlations between SR and thyroid volume as well as body surface area. These parameters have not been evaluated in other studies on the pediatric population so far.

The innovative aspect of our study is the comparison of thyroid parenchyma elasticity between children getting hormone replacement medication and those who are not receiving it. No differences were found between children who receive levothyroxine treatment (1.24 ± 0.26) and those who do not (1.18 ± 0.25; $p = 0.4615$). This direction of research should be further investigated because there are studies on the adult population in which differences were identified between patients who receive treatment and those who do not [52,60]. A study conducted by Magri et al. found significantly higher values of thyroid elasticity using SWE when comparing patients who were on levothyroxine treatment to untreated euthyroid patients (27.3 ± 9.0 kPa vs. 20.9 ± 10.4 kPa, respectively, $p = 0.02$) [60]. Another study using SE found differences between CAT patients who receive treatment and untreated euthyroid patients (3.45 ± 2.53 vs. 2.15 ± 1.27, $p < 0.0001$). A cut-off value >2.94 was found to be predictive for hypothyroid status in patients with CAT [52].

We aim to follow up on these patients to determine the evolution of stiffness and possible correlation with the need for treatment once the disease progresses.

## 5. Conclusions

This paper demonstrated the usefulness of SE in diagnosing CAT in children with very good Sp and Se values (84.62% Se, 95.45% Sp, 97.8% PPV, 72.4% NPV). We did find correlations between antibody titers and elastographic findings (ATPO-r = 0.65; ATG-r = 0.29). The results obtained are important, encouraging, and pave the way for new, wider studies considering the lack of data for this population category. A further study direction could be the investigation of the connection between hormonal substitution treatment and elastographic parameters.

**Author Contributions:** Conceptualization, D.S. and C.M.C.; methodology, A.B. and C.P.; software, R.B.; validation, D.S.; formal analysis, M.S.M. and R.B.; investigation, C.M.C. and C.P.; resources, D.S.; data curation, M.S.M.; writing—original draft preparation, C.M.C.; writing—review and editing, A.B.; visualization, M.S.M.; supervision, D.S.; project administration, C.M.C. All authors have read and agreed to the published version of the manuscript.

**Funding:** This research received no external funding.

**Institutional Review Board Statement:** The study was conducted in accordance with the Declaration of Helsinki, and approved by the local Ethics Committee ("Dr. D Medical Center", CECS nr. 12/20 December 2021). Informed Consent Statement: Informed consent was obtained from all subjects/their parents involved in the study.

**Informed Consent Statement:** Informed consent was obtained from all subjects involved in the study. Written informed consent has been obtained from the patients to publish this paper.

**Data Availability Statement:** The data presented in this study are available on request from the corresponding author. The data are not publicly available due to patient privacy IRB requirement.

**Conflicts of Interest:** The authors declare no conflict of interest.

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
