# Peer review of "Is Strain Elastography Useful in Diagnosing Chronic Autoimmune Thyroiditis in Children?"

_applsci, doi:10.3390/app12178881_

Round 1

Reviewer 1 Report

The study was designed to investigate if Strain Elastography (SE) could be used in diagnosing CAT in children. The study had proper controls (left lobe vs right lobe; CAT patients vs healthy controls; SE vs. conventional US).  The data supported the authors conclusion that SE is a useful examination technique of children with CAT. 

There are several points that the authors need to address:

1.       Although local ethics committee approved the study, it is unclear where did the subjects were recruited, from university affiliated hospitals or from multiple local clinics?

2.       A multivariate regression models was used, however, family history as well as gender were not included in the multivariate regression model. Both factors are known strong risk predictors for CAT.

Author Response

Response to Reviewer 1

Dear Reviewer,

Thank you very much for your relevant comments and generous suggestions. Your review truly did help us improve the manuscript. Also,  grammatical analysis and error correction were done. All changes are highlighted in yellow. Hopefully it will meet your requirements.

  1. Although local ethics committee approved the study, it is unclear where did the subjects were recruited, from university affiliated hospitals or from multiple local clinics?

Thank you for your remark, the subjects were recruited from “Dr D. Medical Center” in Timisoara – (added in the manuscript, chapter 2.2 inclusion criteria).

  1. A multivariate regression models was used, however, family history as well as gender were not included in the multivariate regression model. Both factors are known strong risk predictors for CAT.

Thank you very much for your comment, family history is very important, so we added this into our manuscript (results section). For our group of subjects, gender was not associated with the presence of CAT.

Reviewer 2 Report

This study implemented strain elastography to discriminate various pathologies via measuring the tissue stiffness. The manuscript is well organized and the results are properly presented. There are some minor comments for this manuscript below:

1.    Abstract: The authors only introduce studies done in this manuscript, while they didn’t involve previous studies for diagnosing pathologies. It is important to raise the study significance through comparing previous studies and studies in this work.

2.    Conclusions: It is suggested to presented solid results in conclusions to convince readers, such as giving the sensitivity values, r coefficients in correlations, etc.

3.    There are some grammar mistakes found in the manuscript, the authors should double-check the whole manuscript before submitting again.

Author Response

Response to Reviewer 2

Dear Reviewer,

Thank you for all the constructive comments. This review helped us refine the manuscript and improve the display of our findings. Hopefully it will meet your requirements. All requested changes are highlighted in yellow.

  1. Abstract: The authors only introduce studies done in this manuscript, while they didn’t involve previous studies for diagnosing pathologies. It is important to raise the study significance through comparing previous studies and studies in this work.

Thank you very much for your comment. Since the abstract must contain a limited number of words, we presented the results that we obtained and not other results from the specialized literature. To your suggestion, we added in the abstract that our results are consistent with those found in other studies. (abstract section)

  1. Conclusions: It is suggested to presented solid results in conclusions to convince readers, such as giving the sensitivity values, r coefficients in correlations, etc.

Thank you very much for your suggestion. We added solid result in the conclusion section in order to highlight the importance of our work. (conclusion section)

  1. There are some grammar mistakes found in the manuscript, the authors should double-check the whole manuscript before submitting again.

Thank you for your observation.  Grammatical analysis and error correction were done.

Round 2

Reviewer 1 Report

The authors addressed all the questions.